# A Hybrid Numerical Methodology Based on CFD and Porous Medium for Thermal Performance Evaluation of Gas to Gas Micro Heat Exchanger

**DOI:** 10.3390/mi11020218

**Published:** 2020-02-20

**Authors:** Danish Rehman, Jojomon Joseph, Gian Luca Morini, Michel Delanaye, Juergen Brandner

**Affiliations:** 1Microfluidics Laboratory, Department of Industrial Engineering (DIN), University of Bologna, 40131 Bologna, Italy; gianluca.morini3@unibo.it; 2Institute of Microstructure Technology (IMT), Karlsruhe Institute of Technology, D-76344 Eggenstein-Leopoldshafen, Germany; joseph.jojomon@mitis.be (J.J.); juergen.brandner@kit.edu (J.B.); 3MITIS SA, Rue del Rodje Cinse 98, 4102 Seraing, Belgium; michel.delanaye@mitis.be

**Keywords:** reduced model, LMTD method, conjugate heat transfer (CHT), compressible fluid, maldistribution

## Abstract

In micro heat exchangers, due to the presence of distributing and collecting manifolds as well as hundreds of parallel microchannels, a complete conjugate heat transfer analysis requires a large amount of computational power. Therefore in this study, a novel methodology is developed to model the microchannels as a porous medium where a compressible gas is used as a working fluid. With the help of such a reduced model, a detailed flow analysis through individual microchannels can be avoided by studying the device as a whole at a considerably less computational cost. A micro heat exchanger with 133 parallel microchannels (average hydraulic diameter of 200μm) in both cocurrent and counterflow configurations is investigated in the current study. Hot and cold streams are separated by a stainless-steel partition foil having a thickness of 100μm. Microchannels have a rectangular cross section of 200μm ×200μm with a wall thickness of 100μm in between. As a first step, a numerical study for conjugate heat transfer analysis of microchannels only, without distributing and collecting manifolds is performed. Mass flow inside hot and cold fluid domains is increased such that inlet Reynolds number for both domains remains within the laminar regime. Inertial and viscous coefficients extracted from this study are then utilized to model pressure and temperature trends within the porous medium model. To cater for the density dependence of inertial and viscous coefficients due to the compressible nature of gas flow in microchannels, a modified formulation of Darcy–Forschheimer law is adopted. A complete model of a double layer micro heat exchanger with collecting and distributing manifolds where microchannels are modeled as the porous medium is finally developed and used to estimate the overall heat exchanger effectiveness of the investigated micro heat exchanger. A comparison of computational results using proposed hybrid methodology with previously published experimental results of the same micro heat exchanger showed that adopted methodology can predict the heat exchanger effectiveness within the experimental uncertainty for both cocurrent and counterflow configurations.

## 1. Introduction

Micro heat exchangers (μHx) are becoming of great interest for applications where portability, high energy efficiency and ultra high heat transfer rates are required such as the case with microelectronics cooling. Typically, μHxs are composed of multiple layers (similar to plate type heat exchangers) where each layer contains a large number of parallel microchannels (MCs). Detailed CFD modeling of these devices therefore, requires a significant amount of computational power. This computational burden can be reduced by modeling the MCs as a porous medium. Originally proposed for estimating pressure drop of incompressible flow over a bed of spheres (porous medium), Darcy’s Law has been extended to multiple parallel MCs in heat sinks. An analytical model was first developed by Kim et al. [1,2] where they modeled the MCs as porous media and compared modeling results with experimental results of Tuckerman & Pease [3] as well as with Knight et al. [4]. Results showed that the developed model can be used for thermal performance and optimization of MC heat sinks. The same model has been studied by Liu and Garimella [5] and improved by Lim et al. [6]. Porous media-based analytical models tend to solve a three equations model for fluid flow and heat transfer through MC heat sinks using simplified momentum and energy equations. A porous medium-based computational model was validated against experimental studies of three ceramic μHxs by Alm et al. [7]. They found out that the heat transfer estimated by the porous model is lower than the experimental results and associated it to the channel blockage effects present in the experimental devices that are nearly impossible to cater for, in the numerical model. A common trait of all the studies conducted for MC heat sinks/μHxs is the use of an incompressible fluid. A porous medium approximation of a compact heat exchanger used in a micro gas turbine application has recently been presented by Joseph et al. [8,9,10]. Channel dimensions and operating pressure were such that gas was incompressible whereas the operating temperature of the hot fluid was higher than 1000 K. A porous model approximation for a double layered μHx where gas experiences strong compressibility effects, has recently been presented by the authors [11], for parallel flow arrangement. In the current study, the previously reported work is extended to cover the counterflow arrangement as well and detailed comparisons between experimental and numerical results for both flow configurations are being presented.

In this work, a two step methodology is proposed to conduct a performance evaluation study on μHx with acceptable computational power. A gas to gas double layer μHx as shown in Figure 1 that has been experimentally investigated previously by Yang et al. [12,13] and Gerken et al. [14], is used for validation of proposed methodology.

As a first step, a 3D conjugate heat transfer (CHT) model of gas to gas μHx is developed without distributing and collecting manifolds. Resulting pressure, velocity and temperature fields are utilized to calculate inertial and viscous coefficients of the modified Darcy’s porous law. A complete single layer of μHx with manifolds is then modeled with boundary conditions such that MCs are modeled as a porous medium with a low resolution mesh. This is achieved by modeling the required pressure drop as a momentum source term using inertial and viscous coefficients of the porous medium and applying a free slip boundary condition at all MC walls. Similarly to incorporate heat transfer in MCs core, a source term derived by CHT analysis of MCs only, is also introduced in the complete single layer model.

## 2. Background

Pressure drop (Δp) of a fluid through a porous medium of length *L* can be expressed using an extended Darcy–Forchheimer (here after referred to as simply Darcy) law as follows:(1)$$\frac{-\Delta p}{L}=\frac{\mu \dot{G}}{\alpha \rho }+\frac{\Gamma {\dot{G}}^{2}}{2\rho }$$
where $\frac{1}{\alpha }$ is viscous coefficient representing porous medium permeability and Γ is the inertial coefficient of the Darcy’s law, ρ and μ denote density and dynamic viscosity of the fluid respectively, and $\dot{G}$ denotes mass flow rate ($\dot{m}$) per unit area *A* ($\dot{G}=\dot{m}/A$). Calculation of viscous ($\frac{1}{\alpha }$) and inertial (Γ) coefficients is usually done using experimental pressure drop results and therefore various empirical relations exist for different porous media geometries. No such experimental relations exist for the gas flows in μHx, however. As mentioned earlier, in this work porous medium coefficients are extracted from CHT analysis using a modified Darcy law. Thus, integrating Equation (Equation 1) in the streamwise direction ‘*x*’ of the MC yields:(2)$$\int_{in}^{out}\frac{-\Delta p}{L}dx=\frac{\mu \dot{G}}{\alpha }\int_{in}^{out}\frac{dx}{\rho }+\frac{\Gamma {\dot{G}}^{2}}{2}\int_{in}^{out}\frac{dx}{\rho }$$

Differentiating the ideal gas law resuts in:(3)dp=RTdρ+ρRdT

Utilizing the definition of speed of sound i.e., $\frac{dp}{d\rho }=c^{2}$, above equation can be rewritten as:(4)$$d\rho (c^{2}-RT)=\rho RdT$$

Assuming that temperature change is linear along the length of the MC, following can be obtained:(5)$$dT=\frac{\Delta T}{L}dx$$

Substituting Equation (Equation 5) into Equation (Equation 4):(6)$$dx=\frac{L(c^{2}-RT)}{R\Delta T}[\frac{d\rho }{\rho }]$$

Substituting Equation (Equation 6) into Equation (Equation 2) finally results:(7)$$\frac{-\Delta p}{L}\int_{in}^{out}\frac{d\rho }{\rho }=\frac{\mu \dot{G}}{\alpha }\int_{in}^{out}\frac{d\rho }{\rho^{2}}+\frac{\Gamma {\dot{G}}^{2}}{2}\int_{in}^{out}\frac{d\rho }{\rho^{2}}$$

Integrating Equation (Equation 7) between inlet ‘*in*’ and outlet ‘*out*’ of the MC yields:(8)$$\frac{-\Delta p}{L}=\mu \xi \dot{G}(\frac{1}{\alpha })+\frac{\xi {\dot{G}}^{2}}{2}(\Gamma )$$
where $\xi =\frac{1}{\ln(\frac{\rho_{out}}{\rho_{in}})}(\frac{1}{\rho_{in}}-\frac{1}{\rho_{out}})$.

Boundary conditions of CHT analysis are chosen such that there are no heat losses to the surroundings, therefore all the heat loss by hot fluid must be shared between partition foil (solid wall) and cold fluid. Therefore porous medium coefficients can be extracted from either side of the CHT model. In this work, inertial and viscous coefficients are extracted from the channel with hot fluid. From a CHT analysis of MCs only, all parameters in the Equation (Equation 8) are available for specific mass flow, with inertial and viscous coefficients as the only two unknowns. However, if Equation (Equation 8) is applied to two consecutive mass flows ${\dot{m}}_{i}$ and ${\dot{m}}_{i+1}$, an average value of both the coefficients between these mass flows can be found out by solving a system of two linear equations for two unknowns. This is repeated for the whole range of mass flow being studied and a polynomial fit on these evaluated coefficients is used as an input for the porous model. Similarly, a volumetric heat loss of hot fluid channel is calculated as a function of mass flow from CHT analysis using:(9)$$\dot{q_{v}}=\frac{\dot{m}C_{p}\Delta T}{AL}$$
where $C_{p}$ is the specific heat of fluid at constant pressure. A polynomial fit onto this volumetric heat loss is used as a source term in energy equation while solving for the porous model. Utilizing the results of the CHT model, a MATLAB program is used to implement the above mentioned strategy for the extraction of the porous medium coefficients and the heat source term.

## 3. Numerical Methodology

As described earlier, two different numerical setups are used in the current work. First a 3D CHT of MCs only, without distributing and collecting manifolds is performed. Secondly, using the porous medium coefficients and the heat source term derived from CHT analysis, the performance of the complete heat exchanger with manifolds is analyzed. Reynolds number at the inlet of MC is defined by:(10)$$Re=\frac{\dot{m}D_{h}}{\mu A}$$
where hydraulic diameter ($D_{h}$) of a rectangular MC with width (*w*) and height (*h*) is defined as:(11)$$D_{h}=\frac{2wh}{w+h}$$

From CHT analysis, heat transfer rate ($\dot{Q}$) on hot (h) and cold (c) side can be defined by using respective flow quantities as:(12)$$\dot{Q}=\dot{m}C_{p}\Delta T$$

Finally heat exchanger effectiveness, defined as the ratio of actual heat transfer rate and maximum potential heat transfer rate available, can be calculated using:(13)$$\varepsilon =\frac{Q_{av}}{{\left(\dot{m}C_{p}\right)}_{min}\left(T_{h,in}-T_{c,in}\right)}$$
where ${\dot{Q}}_{av}=\frac{{\dot{Q}}_{c}+{\dot{Q}}_{h}}{2}$, is the average value of the heat transfer evaluated on the cold and hot side of μHx. For porous model of μHx, heat exchanger effectiveness to be compared with experimental results, is calculated for the MC core only. This essentially means that temperature difference from the inlet to outlet of MC core is used to calculate ε. Moreover, as only one layer is computationally modeled in porous model, resulting effectiveness is calculated as follows:(14)$$\varepsilon_{p}=\frac{Q_{p}}{{\left(\dot{m}C_{p}\right)}_{p}\left(T_{h,in}-T_{c,in}\right)}=\frac{{\bar{T}}_{MC,in}-{\bar{T}}_{MC,out}}{T_{h,in}-T_{c,in}}$$
where subscript ‘*p*’ denotes the porous model. ${\bar{T}}_{MC,in}$ and ${\bar{T}}_{MC,out}$ denote the mass flow weighted averages of static temperatures, at the inlet and outlet of *N* number of MCs, respectively. The parameter to quantify mass flow maldistribution from the Porous model is defined as:(15)$${\dot{m}}_{dev}=(\frac{{\dot{m}}_{i}-{\dot{m}}_{ideal}}{{\dot{m}}_{ideal}})\times 100$$
where $m_{i}$ denotes the mass flow through *i*th MC out of total (*N*) MCs. Whereas, ideal mass flow rate (${\dot{m}}_{ideal}$) can be calculated using Equation (Equation 10) for a given Re.

In order to extract inertial and viscous coefficients for the porous medium using methodology outlined earlier, a 3D CHT model is setup where only MCs are modeled without considering both collecting and distributing manifolds. However, to allow any possible underexpansion at the outlet of hot and cold MCs, computational domain is extended $15D_{h}$ in the streamwise and $32D_{h}$ in the lateral direction as shown in Figure 2. A meshed model for a complete layer of μHx is also shown in Figure 3. Geometry and meshing for both models are done using Design Modeler and ANSYS Meshing software respectively. A mesh of 40×40×100 is used in the MCs for CHT analysis whereas a coarse mesh of 3×3×40 is used for MCs in case of porous model. Mesh is refined near the walls to capture any flow vortices present in the model. A commercial solver CFX based on finite volume methods is used for the flow simulations. Ideal Nitrogen gas is used as working fluid for both models. Simulation relevant parameters used in analyses are tabulated in Table 1.

Laminar flow solver is used for the CHT model whereas a transient turbulence model $\gamma -Re_{\theta }$ [16,17,18] is utilized for the porous model. Higher order advection scheme available in CFX is utilized and pseudo time marching is done using a physical timestep of 0.01 s. A convergence criteria of $10^{-6}$ for RMS residuals of governing equations is chosen while monitor points for pressure and velocity at the MC inlet and outlet are also observed during successive iterations. If residuals stayed higher than supplied criteria, the solution is deemed converged when monitor points did not show any variation for 200 consecutive iterations. Reference pressure of 101 kPa was used for the simulation and all the other pressures are defined with respect to this reference pressure. Energy equation was activated using Total energy option available in CFX which adopts energy equation without any simplifications in governing equations solution. Kinematic viscosity dependence on gas temperature is defined using Sutherland’s law. Further details of boundary conditions used in CHT and Porous model can be found in Table 2 and Table 3 respectively.

Using the CHT model, global as well as local evolution of flow variables with inlet mass flow is evaluated at six different cross-sectional planes defined at x/L of 0.005, 0.1, 0.5, 0.9, 0.95 and 0.995 respectively. In addition, two planes defined at x/L of 0.0001 and 0.9995 are treated as the inlet and outlet of MC, respectively. Results from these planes for both hot and cold fluid sides are further post processed in MATLAB to deduce required flow quantities. Thermal effectiveness is then simply evaluated using Equation (Equation 13). Once the porous medium coefficients namely inertial (Γ) and viscous ($\frac{1}{\alpha }$) are determined using CHT model of a double layer μHx, the next step is to setup a complete single layer porous model with inlet and outlet manifolds.

## 4. Results

### 4.1. CHT Model

CHT model with linear periodicity at side walls represents an ideal situation where there exists no maldistribution for parallel MCs. This essentially means that all parallel MCs would have the same mass flow at their respective inlets and manifold does not play a significant role in the performance evaluation. Results for the heat transfer rate for both cocurrent and counterflow configurations are shown in Figure 4. For an incompressible fluid, the heat transfer rate using the numerical model on both sides should be equal. But for gases, as the gas flow experiences additional acceleration due to compressibility, heat gain on the cold side tends to differ from heat loss on the hot side. An interesting fact is that the heat transfer rate on the hot side keeps on increasing with increasing mass flow. On the contrary, it keeps on decreasing on the cold side. This holds for both cocurrent and counterflow configurations. Therefore even though there are no losses modeled to the surroundings in the CHT model, due to compressibility effects gas flows still exhibit a difference in heat transfer rate between hot and cold sides.

Similar behavior can also be seen in effectiveness where it increases with the mass flow for hot fluid and decreases for the cold fluid as shown in Figure 5.

This is simply because gas flow accelerates at the expense of kinetic energy, therefore a part of total energy is utilized for acceleration. Therefore for gas to gas μHxs, it is not recommended to operate in high mass flow regimes because at higher mass flows, gas flows experience a “self cooling” phenomenon at the expense of higher pressure drop. This phenomenon is evident in Figure 6 which shows the evolution of the temperature at the centerline of both fluid streams and partition foil (solid) along the direction of the flow. For smaller mass flows (Re) temperature profiles of hot and cold streams follow the typical trend for incompressible fluids under cocurrent configuration where the temperature of hot and cold fluids is symmetric and partition foil assumes an average constant temperature of the two sides. As fluid velocities increase at the inlets of respective streams, profile symmetry deteriorates with partition foil assuming a temperature that is more influenced by the hot side than the cold fluid side. The outlet temperature of the hot fluid keeps on decreasing with an increased mass flow rate signaling a better heat transfer process. On the contrary, if one is to look at the outlet temperature of the cold fluid a significant decrease is observed by increasing the mass flow rate. This signifies that the cold fluid stream is utilizing the transferred thermal energy from the hot stream, only to increase the velocity (kinetic energy) [19,20,21,22].

Similarly, a higher decrease of temperature and pressure very close to the outlet of the MC for the hot fluid stream is also due to a sudden expansion of the gas due to compressibility. As a result of this strong compressibility effect, gases in both fluid streams are utilizing respective thermal energies and converting them in kinetic energy thereby deteriorating the overall heat transfer process. Temperature decrease of the cold stream is such that the static temperature at the outlet is lower than the inlet value showing no active participation of cold fluid stream into the overall heat exchange of the device at higher mass flow rates. This also explains the decrease of $\dot{Q}$ in Figure 4 on the cold side of the μHx and a continuous increase of $\dot{Q}$ on the hot side with increasing mass flow rates. For data reduction of the most experimental investigations, an average value of $\dot{Q}$ is used to calculate the overall heat transfer coefficient (*U*) to further evaluate the heat exchanger effectiveness (ε). This is done due to the practical limitation of the heat losses to the surroundings in the laboratory environment from both streams. A similar approach has been used by Yang et al. [12,13,15] and Koyama et al. [19,20] for gas to gas μHxs. There are heat losses to the surroundings in a typical experimental campaign but as shown in this work, given a good insulation of the μHx, the real reason for the deviation of $\dot{Q}$ on hot and cold fluid streams in a gas to gas μHx might actually stems from gas compressibility. However, to compare the current numerical results with the experimental results of Yang et al. [13], ε is calculated with an average ${\dot{Q}}_{ave}$ from CHT analysis as it was done in experimental results. The final comparison is shown in Figure 5. For a cocurrent configuration, a CHT analysis overestimates the average ε as compared to experimental results for all the mass flows considered in this study. Difference between the two decreases though at higher mass flow rates. On the contrary, ε for counterflow configuration is underestimated by CHT analysis for the complete range of mass flows investigated. Such discrepancies between experimental results of μHx and an equivalent CHT analysis with periodic boundary conditions are expected as CHT model used in this study is devoid of flow maldistribution effects. Also, any temperature change that might occur in the manifolds due to flow deceleration caused by the presence of numerous circular pillars in distributing and collecting manifolds is not catered for.

Further step is to evaluate the inertial and viscous coefficients of the modified Darcy’s law using flow quantities evaluated in CHT analysis. For this reason, flow characteristics are extracted at various planes along the length of the hot and cold fluid MCs. These are used to form a set of linear systems of equations to solve for porous medium coefficients utilizing the methodology outlined in Section 1. The solutions of these systems of equations for various mass flow rates results in data points where a polynomial with mass flow rate as dependent variable can be fit to be given as input to the porous medium model available in ANSYS CFX^®^. In theory, for the range of inlet gas temperatures considered in this study pressure drop from hot side and cold side should be almost similar and furthermore it should be independent of the flow configuration. Therefore any side of the fluid stream (hot or cold) from CHT analysis can be used to evaluate porous medium coefficients. For the scope of this work, hot fluid side is used to evaluate porous medium coefficients. Resulting viscous and inertial coefficients are shown in Figure 7a,b,d,e for both cocurrent and counterflow configurations.

To model the desired temperature drop on the hot fluid side, a source term $q_{v}$ is extracted from the hot fluid stream of CHT analysis using Equation (Equation 9). The variation of this volumetric heat source term with the mass flow rate is shown in Figure 7c (cocurrent) and Figure 7f (counter flow). A polynomial fit on this $q_{v}$ is given as a source term to energy equation in the porous model of μHx.

### 4.2. Porous Model

The presence of manifolds introduces flow maldistribution that affects both pressure drop as well as heat exchanger effectiveness of a μHx compared to earlier performed CHT analysis. A single layer of the μHx as shown in Figure 3 is considered in the porous model. This is due to the fact that pressure and temperature drops of hot side from CHT analyses that are used to extract porous medium coefficients and source term, respectively, already catered for any conjugate heat transfer effects of both layers. Therefore, modeling only one layer with these coefficients and source term should be sufficient to emulate a double layer μHx. It is important to emphasize at this point that simulating only one layer essentially assumes that the thermal performance evaluated from both hot and cold layers of the μHx will be the same, which does not hold valid for an unbalanced (different mass flow rates on the hot and cold sides) μHx. Therefore, for every change in the geometry of the microchannel and/or unbalanced flow conditions, new coefficients (see Figure 7) are to be attained. Moreover, as porous media coefficients as well as the source term are mass flow rate dependent, based upon encountered maldistribution in the manifold, each MC will exhibit respective pressure and temperature drop along the length. Therefore, contrary to the conventional isotropic porous medium approach to model the MC core as in [1,2,5,6,7], the current porous model behaves as an anisotropic porous medium. This essentially means that each microchannel will exhibit different pressure and temperature drop that enables the observation of the extent of the flow maldistribution.

The experimental pressure drop of the double layer μHx being considered in this study for different flow configurations has been reported by Gerken et al. [14]. Pressure drop showed dependence on the flow configuration as well as the material and thickness of the partition foil employed during the experimental tests. A possible reason for this deviation between different foil materials and thicknesses was associated with possible bending of the thin partition foils inside manifolds although a strong layer of circular pillars was realized underneath (in manifolds) to protect against such undesired deflection of partition foil. Computational results of pressure drop from the current porous model are compared with the experimental results reported for the same μHx from two different studies [13,14] in Figure 8. As results differ from one separation foil material to the other, only results with stainless steel foil with a thickness of 100μm (as utilized in the current study) are compared and are shown in Figure 8a.

It can be seen that the total pressure drop of the device shows a good agreement between the average of two experimental investigations on the same μHx. Results are more compliant with the results of Yang et al. [13] for smaller mass flows while they match better with Gerken et al. [14] for higher mass flow rates. Pressure drops in the distributing manifold and MC core are also shown in Figure 8b where there exists a very good match between the current porous model and experimental results of Gerken et al. [14]. However, the pressure drop of MC core is slightly overestimated at higher mass flow rates with modeled porous medium coefficients with incorporated source term in the energy equation.

Flow maldistribution is shown in Figure 9 for both cocurrent and counterflow configurations where MC indexing is done as outlined in Figure 3a. As expected, maldistribution shows a weak dependence on the flow configuration where except for the furthest MCs from the inlet, it shows similar patterns in both flow configurations. However, due to the presence of circular pillars, maldistribution does not exhibit a typical profile to be expected of triangular manifolds [9,23]. Another reason for this could be the orthogonal direction of the inlet with reference to the channel flow whereas in most experimental as well as numerical studies, an inline flow is encountered where fluid enters parallel to the base plane of distributing manifold.

To compare the effect of heat transfer on the flow maldistribution, counterflow porous model results are calculated at mass flow rate of 2.5 kg/s by deactivating the energy source term and results are shown in Figure 10.

It is evident that temperature drop does not substantially affect the maldistribution pattern in the middle core of the current manifold while it is higher on either extreme for the case when source term is not modeled. In other words, for current configuration heat transfer helps decreasing the maldistribution in first and last MCs.

Heat exchanger effectiveness evaluated using Equation (Equation 14) is shown in Figure 11 for both flow configurations. In the experimental campaign, one pressure and two temperature sensors were placed 0.4 mm from the inlet and outlet of the MCs core to represent an average value of pressure drop and temperature drop or gain through the hot or cold side of MCs core. For a cocurrent configuration, porous model predictions match to the experimental results within the experimental uncertainty. However, for the counterflow configuration, results of the porous model seem to underestimate ε compared to experimental results. To investigate this discrepancy, average fluid temperatures at two planes of 0.5 mm × 0.5 mm are used close to the locations where experimental temperature sensors were placed. The average temperature of the gas is then calculated using a simple equal weighted average of these two temperatures as done in the experimental campaign. As it can be seen in Figure 11b, when temperature estimation at the inlet and outlet of MCs core is done in a way similar to experimental settings, a higher temperature drop and an apparent increase in ε is evidenced. However, when mass flow weighted averages of temperature at inlet and outlet of MC core are considered from numerical simulations, ε is slightly lower than the experimental one. On the other hand, a limited number of sensors are practical limitations inside microdevices therefore, averaging performed using a limited number of temperature and pressure sensors would always result in a slight discrepancy between experimental and numerical global characteristics of the μHxs. Results of ε from the porous model are also compared with the CHT analysis. For counterflow configuration, it can be noted from Figure 11b that porous model in which flow maldistribution is accounted for gives an identical overall heat exchanging efficiency to that obtained with CHT model without manifolds.

A similar conclusion was also presented by Joseph et al. [9] where CHT analysis with periodic side walls for a 60 layered counterflow compact heat exchanger showed only 4% deviation in ε compared to experimental results. This hints that it is sufficient to use a CHT model to predict the thermal effectiveness of a multilayered parallel channel counterflow μHx. Pressure drop analysis of the entire device, however, cannot be performed using only CHT analysis as there is a significant pressure drop in the distributing and collecting manifolds. On the contrary, ε for the cocurrent configuration shows a dependence on the maldistribution, in fact CHT model overestimates ε when compared to porous model (see Figure 11a).

## 5. Conclusions

Independent of the flow configurations investigated in this study, the developed porous medium-based numerical model shows an excellent match with experimental results for both cocurrent and counterflow configurations. Moreover, as flow inside porous MCs is not treated as wall bounded, rather a free slip boundary condition is applied, a substantial amount of computational power is saved in the process. Since the proposed porous medium model requires only one layer of μHx, with MCs core modeled as porous medium, it eliminates the need of simulating all the layers of such a microdevice where a layered structure is quite common and an effort to computationally model such a device with all the layers would require a staggering amount of computational power. Therefore, the proposed model for the analysis of parallel channel μHx can prove to be a feasible option for rapid design optimization in engineering studies. Based on the discussion conducted earlier, the following conclusions can be inferred:Porous medium coefficients for parallel channel μHx can be extracted for compressible fluids by modifying the existing Darcy–Forchheimer law to incorporate for the strong density variations with increasing mass flow rates in MCs.CHT analysis revealed that gas in both hot and cold fluid streams experiences a “self-cooling” phenomenon where the temperature of the gases keeps on decreasing from inlet to the outlet at higher mass flow rates (Re). Therefore for a μHx operating under balanced mass flow rates, smaller values of mass flow rates are recommended.Pressure drop of the porous model is much higher compared to the CHT due to the presence of collecting and dividing manifolds. Pressure drop estimation using the porous model is in good agreement with the experimental results of the same μHx.Overall heat exchanger effectiveness of a μHx in a counterflow configuration is identical to that of the CHT analysis on the range of mass flows investigated in this study. For a cocurrent configuration, however, heat exchanger effectiveness from porous model matches well with experimental results while the CHT model overpredicts it.Compared to the meshing strategy adopted in CHT analysis, the porous model results in saving of at least 20 million computational nodes for the double layer gas to gas μHx investigated in this study with good enough predictions of global pressure drop and heat exchanger effectiveness.

## Figures and Tables

**Figure 1 micromachines-11-00218-f001:**
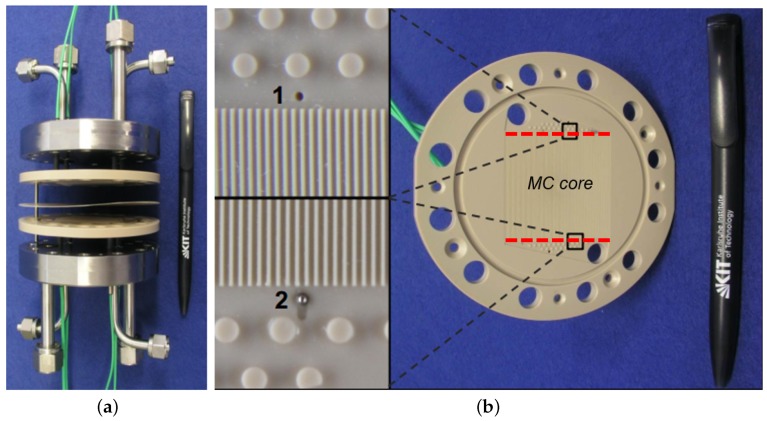
Experimental assembly for double layer Micro heat exchangers (μHx) (taken from [15]) (**a**), and zoomed view of single layer (**b**).

**Figure 2 micromachines-11-00218-f002:**
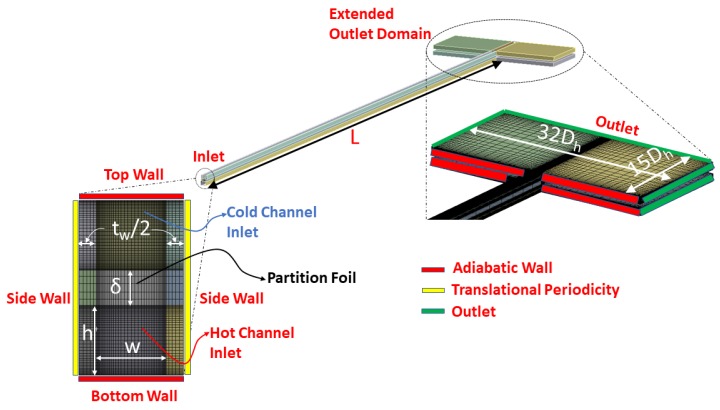
Mesh and geometric details for co-current conjugate heat transfer (CHT) analysis.

**Figure 3 micromachines-11-00218-f003:**
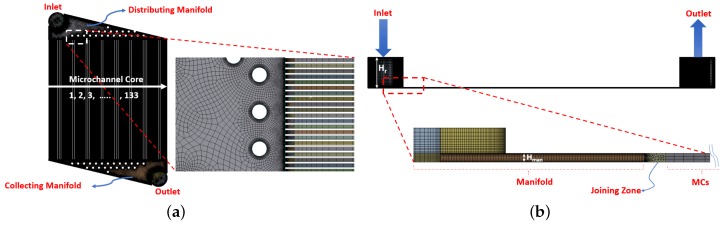
Geometry and mesh details of porous model: top view (**a**), and side view (**b**).

**Figure 4 micromachines-11-00218-f004:**
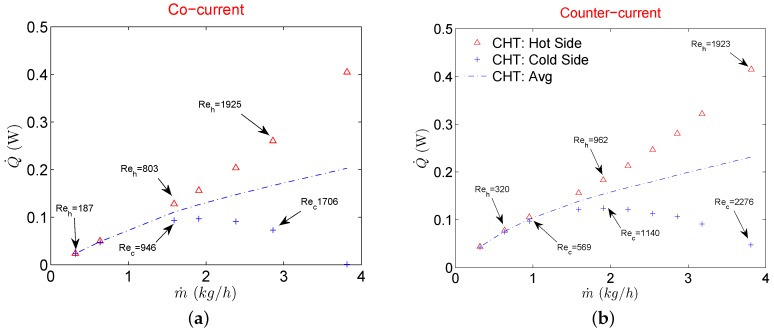
Heat transfer rate for CHT analysis when flow configuration is cocurrent (**a**), and counterflow (**b**).

**Figure 5 micromachines-11-00218-f005:**
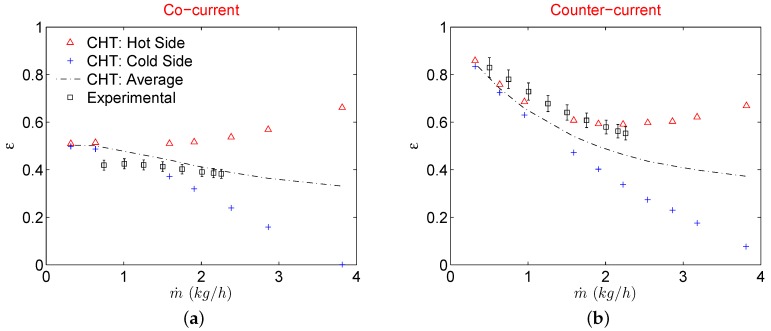
Heat exchanger effectiveness for CHT analysis when flow configuration is cocurrent (**a**), and counterflow (**b**).

**Figure 6 micromachines-11-00218-f006:**
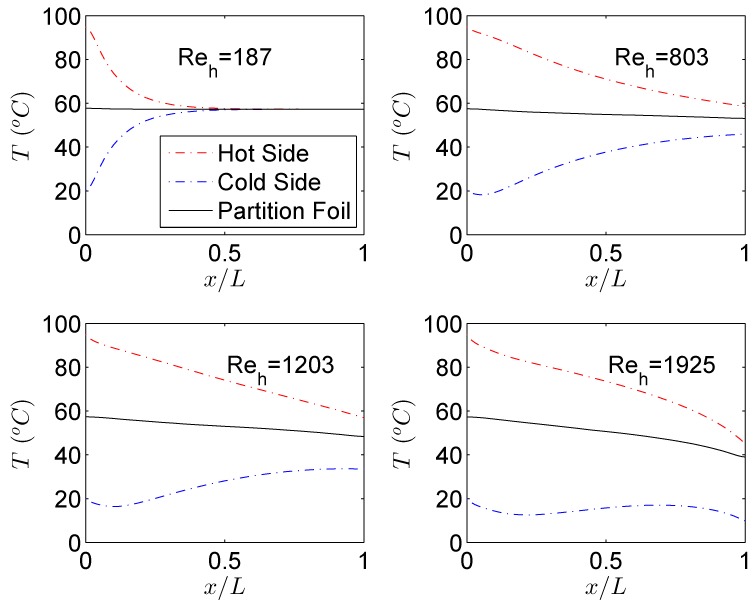
Temperature along the length of the hot and cold MCs for various Re in cocurrent flow configuration.

**Figure 7 micromachines-11-00218-f007:**
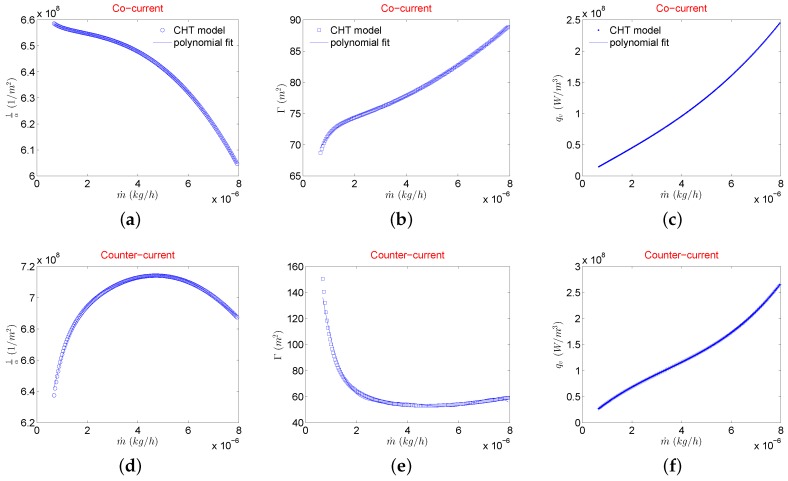
Viscous coefficient (**a**,**d**), inertial coefficient (**b**,**e**), and volumetric heat source term (**c**,**f**) extracted from CHT analysis in both flow configurations.

**Figure 8 micromachines-11-00218-f008:**
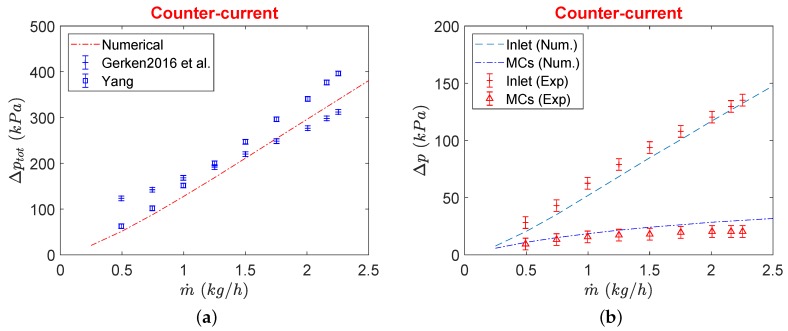
Comparison between experimental and numerical total pressure drop of μHx (**a**), and in the inlet and MCs only (**b**).

**Figure 9 micromachines-11-00218-f009:**
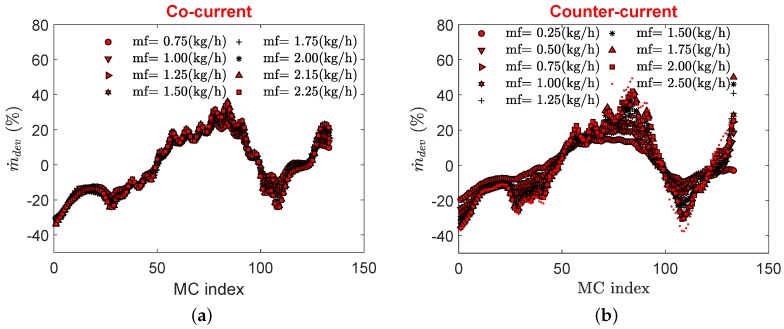
Flow maldistribution in MCs for cocurrent flow (**a**), and counterflow (**b**).

**Figure 10 micromachines-11-00218-f010:**
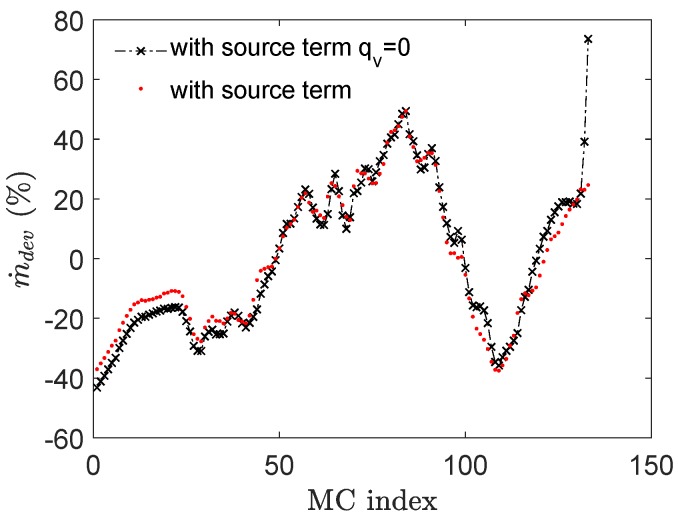
Flow maldistribution with and without source term $q_{v}$ for counterflow configuration with ${\dot{m}}_{f}$ = 2.5 kg/s.

**Figure 11 micromachines-11-00218-f011:**
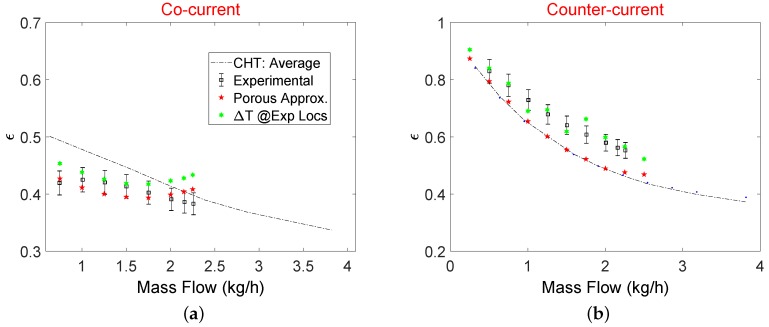
μHx effectiveness for cocurrent flow (**a**), and counterflow (**b**).

**Table 1 micromachines-11-00218-t001:** μHx geometry used for simulations.

Parameter	Symbol (Units)	Value
MC width	w(μm)	200
MC height	h(μm)	200
MC Length	L(μm)	40
Hydraulic Diameter	$D_{h}\;(\mu$m)	200
Wall Thickness	$t_{w}\;(\mu$m)	100
MC housing (PMMA) conductivity	$k_{MC}$ (W/m/K)	0.25
Partition Foil (Stainless Steel) thickness	δ(μm)	100
Partition Foil conductivity	$k_{PF}$ (W/m/K)	15

**Table 2 micromachines-11-00218-t002:** Boundary conditions used in the CHT Analysis.

Boundary	Value	
	Hot Side	Cold Side
**Inlet**	$-\dot{m}$ evaluated using Equation (Equation 10) for cold side	
	$-T_{h,in}=90$${}^{\circ }$C	$-T_{c,in}=20$ ${}^{\circ }$C
**Side Walls**	Translational Periodicity	
**Top & Bottom Walls**	Adiabatic	
**Outlet**	Pressure outlet, p = $p_{atm}$	

**Table 3 micromachines-11-00218-t003:** Boundary conditions used in the porous model for μHx.

Boundary	Value
**Inlet**	$-\dot{m}$ from experimental testing
	$-T_{h,in}$ = 90 ${}^{\circ }$C
**MCs walls**	Free slip
**Inertial and visocus coefficients**	Determined from CHT analysis
**Energy source term**	Determined from CHT analysis
**Manifolds walls**	Adiabatic/ No slip
**Outlet**	Pressure outlet, p = $p_{atm}$

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
