# Peer review of "A Hybrid Numerical Methodology Based on CFD and Porous Medium for Thermal Performance Evaluation of Gas to Gas Micro Heat Exchanger"

_micromachines, 2020, doi:10.3390/mi11020218_

Round 1

Reviewer 1 Report

The paper describes a novel methodology that can be employed to evaluate the thermal performance of gas to gas micro heat exchangers. According to the proposed approach, the microchannels are modelled as a porous medium where a compressible gas is used as the working fluid. The co-current and counterflow configurations are considered. The inertial and viscous coefficients of the porous medium necessary to account for the pressure drop in the microchannels of the heat exchanger are obtained from a preliminary 3D conjugate heat transfer (CHT) analysis concerning the flow in just two stacked microchannels. The numerical results are compared to literature experimental data concerning a micro heat exchanger with 133 parallel microchannels.

The topic of the article falls within the scope of the journal and is of interest since the proposed methodology can expedite the evaluation of the thermal performance of gas to gas micro heat exchangers reducing the time and the computational effort compared with what would be required by conventional CFD. The successful comparisons of the numerical and experimental results confirm the validity of the proposed approach. The English is good, except for the article “the” being systematically omitted where, instead, it would be required. In the following, a few minor remarks are reported, mainly aimed at improving the readability of the paper.  

Page 3. The procedure for obtaining the coefficient “zeta” (right below Eq. 3) is not straightforward. Since this is a relevant feature of the proposed model, the details of its derivation should be reported. Page 5. The descriptions of the CHT and porous models are interweaved. This makes things a bit confusing. Separate descriptions, one after the other, with the specification of the equations that are actually solved within each model, would make everything clearer. Additional questions concerning the models. (i) What is the range of allowable Mach numbers? (ii) Are the parameters depicted in Fig. 7 only valid when the mass flow rates in the upper and lower microchannels are the same? (iii) In the porous model, are the values of the parameters of Fig. 7 assumed constant in each microchannel, but different for different microchannels on the basis of the mass flow rate, regardless of the temperature distributions? If so, the above assumptions should be clearly stated and discussed. Page 10. The sentence “Contrary to the intuition, maldistribution shows dependence on the flow configuration…” points out a problem, but no explanation is given for the discrepancies. The subsequent text on the same page attributes the somehow unexpected velocity profiles to the presence of the pillars, but it does not justify the differences yielded by the same manifolds with co-current or counter-current flows. Duplicate symbols. The same symbol “A” is used in Eqs. 4 and 5 to indicate an area which is different from that corresponding to the symbol “A” appearing in Eq. 8. Different symbols should be used. Moreover, a precise definition of the heat transfer area in Eq. 8 should be given. The same applies to the symbol “Delta T”, which is used in Eqs. 4, 7 and 8 with different meanings. The style of the figures is not consistent: Some of the plots are contained in rectangles and some are not (only the horizontal and vertical axes are shown); the size of the lettering varies from figure to figure. For consistency, the numerical results should be shown using a dashed line also in Fig. 8(b).

Author Response

We would like to thank the reviewers for their deep review of the paper and for their suggestions and comments that we have found very useful and pertinent indeed. We have prepared the new version of the paper by considering all the suggestions made by the reviewers. You can find below the detailed response to the main questions of the reviewers.

Reveiwer-1

Q1.1    Page 3. The procedure for obtaining the coefficient “zeta” (right below Eq. 3) is not straightforward. Since this is a relevant feature of the proposed model, the details of its derivation should be reported.

A1.1    Equations (3) to (7) have been added in the new version of the paper (see page 3) to explain the derivation of the coefficient “zeta”.

Q1.2    Page 5. The descriptions of the CHT and porous models are interweaved. This makes things a bit confusing. Separate descriptions, one after the other, with the specification of the equations that are actually solved within each model, would make everything clearer.

A1.2    The general workflow for the proposed methodology is added to the Introduction section (page4;1st paragraph):

A 3D CHT analysis is first performed and using the obtained values of the flow variables a MATLAB code is used to solve the linear system of equations (Eq. (8) by varying the mass flow rate within the range investigated). The solution of this system of equations gives the inertial and viscous coefficients (see Fig. 7). These coefficients are then used in the porous model to estimate the thermal performance of a single layer (hot fluid side in this case) of the investigated micro heat exchanger.

Due to the reason that both models (CHT and porous) are implemented in CFX solver, it is easier to describe the solver and mesh settings together instead of repeating the same sentences for both models, separately. Moreover, the post processing of the results is also performed in a similar fashion for both models and therefore it is mentioned together.

Q1.3    Page 10. The sentence “Contrary to the intuition, maldistribution shows dependence on the flow configuration…” points out a problem, but no explanation is given for the discrepancies. The subsequent text on the same page attributes the somehow unexpected velocity profiles to the presence of the pillars, but it does not justify the differences yielded by the same manifolds with co-current or counter-current flows.

A1.3    There was a misprint. A comparison of the extent of the maldistribution highlights out that it does not vary significantly from one flow configuration to the other. Therefore, previous statement has been modified (lines 209-211).

Presence of pillars on the other hand, only causes an atypical maldistribution profile when compared to the maldistribution trends of triangular manifolds found in the open literature. The reason for such difference is associated with the presence of pillar and the inlet direction of the working fluid which is orthogonal in the current case and is generally parallel for large part of the studies presented in the open literature.

Q1.4    Duplicate symbols. The same symbol “A” is used in Eqs. 4 and 5 to indicate an area which is different from that corresponding to the symbol “A” appearing in Eq. 8. Different symbols should be used. Moreover, a precise definition of the heat transfer area in Eq. 8 should be given. The same applies to the symbol “Delta T”, which is used in Eqs. 4, 7 and 8 with different meanings.

A1.4    Equation (5) in the previous version has been removed because the overall heat transfer coefficient U is not discussed in the current paper. Thus, the need of defining heat transfer area is no more needed. “Delta T” on the other hand, does not need to be redefined as it essentially is the temperature difference between inlet and outlet of the microchannel either on the hot side or cold side in all the paper.

Q1.5    The style of the figures is not consistent: Some of the plots are contained in rectangles and some are not (only the horizontal and vertical axes are shown); the size of the lettering varies from figure to figure. For consistency, the numerical results should be shown using a dashed line also in Fig. 8(b).

A1.5    All the figures are updated as advised.

Q1.6    Additional questions concerning the models.

  1. What is the range of allowable Mach numbers?

 Modeling technique does not restrict the allowable Mach number. However, in our case the maximum value reached by the Mach number was ~0.5 at the outlet.

  1. Are the parameters depicted in Fig. 7 only valid when the mass flow rates in the upper and lower microchannels are the same?

Yes, viscous and inertial coefficients, as well as the temperature sink term, presented in Fig. 7 are only valid for the balanced flow conditions. For an unbalanced heat exchanger, the heat transfer rate will change, and consequently different temperature drops in the hot and cold fluid streams are expected. Thus, all the coefficients would be different in both hot and the cold sides of the heat exchanger. This essentially means that in case of unbalanced flows, porous model must have 2 layers (one for each fluid side) where each layer is modeled with the respective coefficients. This has been highlighted in lines 178-183 of the revised manuscript.

  • In the porous model, are the values of the parameters of Fig. 7 assumed constant in each microchannel, but different for different microchannels on the basis of the mass flow rate, regardless of the temperature distributions? If so, the above assumptions should be clearly stated and discussed.

Yes, in the porous model, values of the coefficients change from one microchannel to the other based upon the mass flow. Similarly, temperature sink term is also mass flow rate dependent and changes for each of the microchannels. Therefore, based on the encountered pattern of the maldistribution in the manifold, the pressure loss and temperature drop/gain of each channel will be different which is amongst the novelties of the porous medium approach. A further clarification of this has been added in lines 183-189 to better highlight the novelty of this approach.

Reviewer 2 Report

Presented methodology is interesting but not new. Similar approach may be found in a number of HE numerical studies basically because of very low numerical costs. However, I haven't seen an implementation to micro-HE yet.
Many details of the phenomena has been taken into account. The free slip BC is correctly chosen and overall quality of numerical simulations is good. However, a few aspects of presentation and interpretation of results should be addressed.

Figures 4 and 5 are unclear. What are the other conditions for heat transfer? How is hot-cold side linked in this figure? Heat flux depends on both side heat transfer conditions, why and how is it analyzed separately?

Figure 8 has no error bars. The discrepancy between experimental results should be addressed. Which of the experimental data is more reliable for validation of the numerical model?

CHT model is based on real geometry, which automatically blocks y,z components of velocity. How about porous model? Is it anisotropic?

5 out of 22 references is recent. The score could be higher taking into account the fact that the micro-Hx is an up-to-date topic of interest.

Author Response

We would like to thank the reviewers for their deep review of the paper and for their suggestions and comments that we have found very useful and pertinent indeed. We have prepared the new version of the paper by considering all the suggestions made by the reviewers. You can find below the detailed response to the main questions of the reviewers.

Reviewer-2

Q2.1    Figures 4 and 5 are unclear. What are the other conditions for heat transfer? How is hot-cold side linked in this figure? Heat flux depends on both side heat transfer conditions, why and how is it analyzed separately?

A2.1    Figs. 4 and 5 represent the heat power and thermal efficiency of the CHT model which are calculated using Eqs. (12) and (13), respectively. In heat exchangers operating with an incompressible fluid, heat loss of hot fluid equates to the heat gain of the cold fluid. Therefore, one single value i.e. Qav which is defined after Eq. (13), is generally used to represent the overall heat power of the heat exchanger. However, heat gain on the cold side (Qc) and heat loss on the hot side (Qh), tend to differ when a compressible gas is used as the working fluid due to the fluid compressibility. This is evident in Fig. 4 which also causes a difference in the evaluation of thermal efficiency in Fig. 5. However, in the paper an average value has been utilized in order to compare the results with experimental data obtained as average values between the cold and the hot side.

Q2.2    Figure 8 has no error bars. The discrepancy between experimental results should be addressed. Which of the experimental data is more reliable for validation of the numerical model?

A2.2    Fig.8 has been updated with the error bars.

Q2.3    CHT model is based on real geometry, which automatically blocks y,z components of velocity. How about porous model? Is it anisotropic?

A2.3    CHT analysis is based on the real geometry and therefore as with the pressure driven flow in a single duct, resulting velocity components in y and z directions are negligible. However, in the porous model, the manifold introduces flow maldistribution i.e. mass flow rate changes at the entrance of each of the 133 microchannels. Since microchannels are not modeled as one black box (conventional isotropic porous model in the literature) rather each MC is modeled individually, this results in a different pressure as well as temperature drop/gain in each microchannel. Therefore, the proposed methodology is anisotropic which means temperature and pressure drops are not uniform in all the microchannels. This has been highlighted in lines 183-189 of the revised manuscript.

Q2.4    5 out of 22 references are recent. The score could be higher taking into account the fact that the micro-Hx is an up-to-date topic of interest.

A2.4    Although micro heat exchangers are of great interest, studies where gas is used as the working medium are still scarce in the open literature. In this work, relevant findings from double/multi layered gas to gas micro heat exchangers are emphasized rather than liquid micro heat exchangers. Currently reported methodology is novel in regards that it provides viscous and inertial coefficients of Darcy law that are valid for a highly compressible fluids such as the Nitrogen gas in current case. For the abundant literature of micro heat exchangers with liquid flows therefore, current methodology to evaluate inertial and viscous coefficients is not valid and hence they can be modeled with conventional Darcy law (isotropic), as discussed in the open literature.
